# Measuring Perceived Voice Disorders and Quality of Life among Female University Teaching Faculty

**DOI:** 10.3390/jpm13111568

**Published:** 2023-11-01

**Authors:** Nisreen Naser Al Awaji, Khaled Abdulraheem Alghamdi, Abdullah Mohammed Alfaris, Rahaf Zamil Alzamil, Lojain Naser Alhijji, Ghaida Saad Alyehya, Shadan Mohammed Al Harbi, Eman M. Mortada

**Affiliations:** 1Department of Health Communication Sciences, College of Health and Rehabilitation Sciences, Princess Nourah bint Abdulrahman University, P.O. Box 84428, Riyadh 11671, Saudi Arabia; nnalawaji@pnu.edu.sa (N.N.A.A.); 441002619@pnu.edu.sa (R.Z.A.); 441000836@pnu.edu.sa (L.N.A.); 441001261@pnu.edu.sa (G.S.A.); 441001832@pnu.edu.sa (S.M.A.H.); 2Rehabilitation Department, King Abdullah Bin Abdulaziz University Hospital, P.O. Box 47330, Riyadh 11552, Saudi Arabia; kalghamdi@kaauh.edu.sa (K.A.A.); amalfaris@kaauh.edu.sa (A.M.A.); 3Health Sciences Department, College of Health and Rehabilitation Sciences, Princess Nourah bint Abdulrahman University, P.O. Box 84428, Riyadh 11671, Saudi Arabia

**Keywords:** teaching faculty, females, quality of life, self-perceived voice disorders

## Abstract

Background: Occupations that require heavy vocal use can place the person at risk of voice disorders (VDs). Heavy demands on the voice, especially for a long time or with loud back-ground noise, can lead to vocal abuse or misuse. The study aimed to measure the prevalence of perceived voice disorders among the teaching faculty at a female university, identify the risk fac-tors that affect their voice, and determine the effect of perceived voice disorders on their quality of life (QoL). Methods: The study sample consisted of female teaching faculty (N = 401). The ques-tionnaire included general sociodemographic data, general voice data, the vocal tract discomfort (VTD) scale, and the World Health Organization Quality of Life assessment (WHOQOL)-BREF. Results: The results demonstrated that 44.1% of the participants had perceived voice disorders, and stress, reflux, and asthma had a significant relationship with self-perceived voice disorders. Furthermore, the data showed that self-perceived voice disorders negatively impacted the overall QoL of teaching faculty. Conclusions: Perceived voice disorders are affected by various factors, including health conditions, medications, and lifestyle choices. Although teaching characteristics and demo-graphic factors are believed to be the cause, in this study they did not significantly contribute to perceived voice disorders. Faculty members with perceived voice disorders have a poorer quality of life, highlighting the need for education on preventative vocal measures and awareness of voice care.

## 1. Introduction

Some occupations require employees to rely heavily on their voices. It has been estimated that about one third of the global workforce is employed in professions in which the voice is the primary instrument [1]. University teachers and schoolteachers are among the heaviest vocal users and have been classified as professional voice users [2,3]. Teachers are at the center of much research that reports their higher risk of voice disorders (VDs) compared to other professions (e.g., [4,5]). Despite the similar classification of teachers and university teachers as nonprofessional voice users, their job roles differ in the ages of the students, the teaching environment, research activities, and their administrative roles [6,7,8]. Therefore, university teachers should be studied as a distinct group. According to the American Speech-Language-Hearing Association (ASHA), VDs are described as “the abnormal production and/or absence of vocal quality, pitch, loudness, resonance, and/or duration, which is inappropriate for an individual’s age and/or gender” [9]. Heavy vocal use for a long duration at a high intensity is often associated with VDs due to changes in the larynx (e.g., muscle fatigue and muscle tension) or the occurrence of vocal pathologies (e.g., polyps and nodules) [10]. VD can also occur as a consequence of direct or indirect medical and health conditions. For example, chronic sinusitis and upper respiratory infections may contribute indirectly to the development of dysphonia. This is because sinus drainage does not touch the vocal folds, but the coughing and throat clearing accompanied by drainage directly cause dysphonia. Other common infections, such as colds and influenza, can lead to irritation and inflammation of the vocal folds and larynx structures, thus causing VDs [10]. Such common infections may thus place faculty individuals at a higher risk of developing VDs. A systematic review by Azari et al. revealed that around 41% of university teachers have a voice disorder [11]. Caffeine consumption was the most common related factor, whereas a dry throat was the most common symptom.

A study on the prevalence and risk factors of VD among university teaching faculty and its effects on their quality of life (QoL) is needed to shed light on the incidence of VDs among teaching faculty, which could negatively impact their teaching abilities and job satisfaction. Second, the identification of risk factors associated with VDs can help prevent or reduce their development. Lastly, the findings about VDs and their effect on the overall QoL of teaching faculty can increase awareness of the importance of vocal health and provide reassurance for faculty members with VDs.

The study objective was to measure the prevalence of self-perceived voice disorders among university teaching faculty, identify the risk factors that might affect their voices, examine their attitudes towards self-perceived VDs, and determine the effect of voice problems on their quality of life.

## 2. Methods

### 2.1. Study Design and Setting

A Cross-Sectional Study Was Conducted among Female Teaching Faculty.

### 2.2. Sample Size and Sampling Techniques

The study sample consisted of 401 female teaching faculty from a governmental university in Saudi Arabia using an open-source statistical package, namely OpenEpi (https://www.openepi.com/Menu/OE_Menu.htm, accessed on 6 October 2023).

### 2.3. Study Tool

A self-reporting online questionnaire was distributed electronically to the partici-pants through email, social media, and other communication channels. The questionnaire was divided into four main categories:

The vocal tract discomfort (VTD) scale [12] is a self-rating scale used for subjective evaluation of voice disorders, that enables occupational voice users to quantify their per-ception of vocal tract discomfort. It is a valuable perceptual indicator of sensory changes in the vocal tract that can be used in the assessment of voice disorders, particularly in its early stages. This scale measures the frequency and severity of symptoms of vocal discom-fort, including burning, tightness, dryness, aching, tickling, soreness, irritability, and a lump in the throat. Frequency was rated on a six-point Likert scale: (never = 0, sometimes = 2, often = 4, and always = 4) with a total score ranging from 0 to 48. Severity was rated on a 6-point Likert scale: (none = 0, mild = 2, moderate = 4, and severe = 6) with total scores ranging from 0 to 48. The participants were divided into two groups using the cut-off scores for the VTDS was: 23. Those with a VTD scale score > 23 were included in the first group with self-perceived voice disorders (PVD), and the other group included people without perceived voice disorders (NPVD) with a NPVD scale score ≤ 23 [13]. 

The risk factors for self-perceived voice disorders that were investigated included the following: (i) personal and professional factors, such as age, marital status, academic rank, and the college the faculty worked in; (ii) teaching factors, such as administrative posi-tions with teaching duties, the average number of teaching hours (hours/week), class size (average number of students/class), and whether the participant was involved in remote teaching; (iii) health and behavioral factors, including smoking status and the amount of water, caffeinated coffee and tea, and decaffeinated coffee and tea drunk/day, and if prac-ticing other hobbies using their voice; and (iv) history of medical conditions reported by the participant and medications to treat the conditions.

The World Health Organization Quality of Life-BREF (WHOQOL–BREF) assessment was used because QoL is considered to be related to various objective and subjective measures. Objectively, a person’s physical and mental health partly defines QoL. On the other hand, subjectively, many factors, which differ from person to person, contribute to measuring QoL, such as the environment, self-fulfillment, and a fully functioning social life [14]. This scale is a self-administered, generic questionnaire that contains 26 items from different domains measuring physical health, psychological health, social relations, and the environment. The standardized Arabic version of the World Health Organization Quality of Life (WHOQOL-BREF) assessment was used to measure the overall QoL among the teaching faculty with voice problems, and Awadalla found high reliability, validity, good internal consistency (alpha = 0.92), and highly significant structural integrity char-acteristics [15].

### 2.4. Ethical Considerations

The research received ethical approval from the Institutional Review Board at Princess Nourah bint Abdulrahman University (IRB: 22-0661), Riyadh, Saudi Arabia. The survey began with a brief explanation of the study and the supervisor’s contact details. To obtain consent from the participants, a question was included in the survey to ensure their willingness to participate in the study. 

### 2.5. Statistical Analysis 

The collected data were analyzed using SPSS v. 22. Descriptive statistics of measures of central tendency and dispersion were calculated. The Mann–Whitney test, independent Student *t*-test, Fisher’s exact test, and the Chi-square test were used to measure the associations between study variables with an odds ratio (OR) and 95% confidence interval (CI) to measure the strength of the associations. A *p*-value ≤ 0.05 was considered statistically significant. 

## 3. Results

The respondents were divided into two groups: 177 (44.1%) participants were in the PVD group, and 224 (55.9%) were in the NPVD group. The sociodemographic profiles of the study groups are portrayed in Table 1; most (43.4%) were aged ≤39 years. Almost half (52.1%) had three or more children, and the majority (76.6%) were married. Regarding professional characteristics, most of the participants were assistant professors and work-ing in humanities colleges (34.2% and 39.9%, respectively).

Table 1 also illustrates the results of the Chi-square test used to assess the relation-ship between self-perceived voice disorders and the participants’ sociodemographic and professional characteristics. No significant differences were found between the study groups regarding their sociodemographic and professional characteristics (*p*-values > 0.05). The English teachers and demonstrators showed voice problems two times as often compared to the other participants (OR 1.87, 95% CI 0.63–5.60 and OR 1.75, 95% CI 0.67–4.62, *p* = 0.56, respectively).

On assessing the relationship between perceived voice disorder and the participants’ work-related characteristics, no significant associations were detected between either study group and all work-related characteristics (*p*-values > 0.05). However, the participants who taught fewer than 15 h per week showed an approximately two times higher risk of perceived voice disorders compared to the other participants (OR 1.71, 95% CI 0.854–3.42, *p* = 0.22) (Table 2). Similar results were found for the university staff members who had an average of 41–50 students per class and fewer than 51 students per class, who were about twice as likely to experience voice disorders as the other staff members (OR 1.56, 95% CI 0.698–3.48 and OR 1.60, 95% CI 0.78–3.12, *p* = 0.13, respectively) (Table 2).

The results for measuring the relationship between perceived voice disorders and health conditions and if it is treated or not are displayed in Table 3 and Table 4, whereas the relationship between perceived voice disorders and health conditions revealed statistically significant associations between perceived voice problems and most of the investigated health conditions (*p*-values ≤ 0.05). Voice problems were almost four times more common among those suffering from hearing impairment and fatigue (OR 3.83, 95% CI 1.66–8.83, *p* = 0.002 and OR 3.54, 95% CI 2.28–35.50, *p* < 0.001, respectively). Similarly, voice problems were nearly three times more prevalent among staff suffering from reflux (OR 2.74, 95% CI 1.78–4.23, *p* < 0.001) and around twice as common among the staff suffering from allergies (OR 2.42, 95% CI 1.55–3.76), depression (OR 2.29, 95% CI 1.25–4.21, *p* = 0.007), and frequent common colds (OR 2.10, 95% CI 1.30–3.39, *p* < 0.001) (Table 3). On the other hand, on assessing the relationship between perceived voice problems and the treatment of the previously mentioned health conditions in Table 3, the results revealed that only two of the treated health conditions, namely allergies and reflux, were statistically significantly associated with perceived voice problems (*p*-values ≤ 0.05). Voice disorders were nearly two and a half times as common among those receiving treatment for allergies and reflux (OR 2.44, 95% CI 1.37–4.34, *p* = 0.002 and OR 1.59, 95% CI 0.76–3.36, *p* = 0.001, respectively) (Table 4).

Table 5 shows the relationship between voice problems and lifestyles. Voice problems were about 29 times more common among staff consuming more than six cups of caffeinated coffee and tea per day (OR 28.77, 95% CI 4.35–190.28, *p* < 0.001) and two times more common among current smokers and those practicing other hobbies using their voice (OR 2.13, 95% CI 0.50–9.04, *p* = 0.58 and OR 1.52, 95% CI 0.93–2.49, *p* = 0.09, respectively).

Figure 1 portrays the prevalence of voice problems among the faculty members: dryness in the throat was the most frequently reported voice symptom (83.3%) and soreness was the least frequently reported (15.2%). The Chi-square test that compared the frequency of the voice-related symptoms between both study groups showed that all the symptoms were significantly higher in those with voice problems (*p*-values < 0.001).

The most severe symptom experienced by the participants was dryness in the throat (81.5%), with tickling being the least severe reported symptom (14.0%). The severity of the voice-related symptoms was significantly higher in those with perceived voice disorders (*p*-values < 0.001), as shown in Figure 2.

Regarding the attitudes of the teaching faculty towards teaching, the overall mean attitude score was 8.20 ± 2.4 out of 15, resulting in 54.7% of respondents who thought that teaching was the reason for PVD (Table 6). The Student’s *t*-test revealed that the mean attitude scores differed significantly between the study groups. Those participants with voice problems had significantly higher mean scores and thought that teaching may cause voice disorders (3.17 ± 1.004 vs. 3.88 ± 0.92, t (399) = −7.296, *p* < 0.001). Likewise, the mean scores of those who thought that they had a disorder with their voice as a result of teaching and those who previously had taken time off work due to a voice disorder significantly differed between the study groups (2.40 ± 1.084 vs. 3.25 ± 1.10, t (399) = −7.749 and 1.79 ± 0.86 vs. 2.15 ± 1.09, t (399) = −3.712, *p* < 0.001, respectively).

On assessing the quality of life of the teaching faculty, the mean scores were significantly lower in those experiencing perceived voice disorders regarding all QoL subscales (*p*-values < 0.05). The median scores significantly differed between the respondents regarding how they rated their quality of life and how satisfied they were with their health (*U* = 1432.5, *Z* = −5.10, and U = 1470.3, Z = −4.81, *p* < 0.001, respectively). The Student’s *t*-test demonstrated that the mean scores of the WHOQOL-BREF were significantly lower in those experiencing voice disorders (54.43 ± 10.32 vs. 61.22 ± 8.91, t (399) = −6.17, *p* < 0.001) compared to those not experiencing voice disorders (Table 6).

## 4. Discussion

Teaching has been recognized as an occupation at high risk for VD [16,17,18,19]. An example is a study that compared subjective voice activity limitation and participation restriction based on the level of professional voice use after controlling for benign vocal disease, gender, and severity using the VAPP questionnaire. According to the study results, elite vocal performers experienced more subjective voice problems related to their jobs than non-professionals, while professional voice users, including teachers and professors, also reported more significant voice problems related to social communication and employment [18].

The present study investigated the prevalence of PVDs among teaching faculty at a female university, the risk factors that affected their voice, and their attitudes toward PVDs. It evaluated the impact of voice problems on their quality of life.

Out of our study population of 401 university faculty members, 44.1% had PVDs, which is in keeping with a study conducted by Higgins and Smith [20]. The faculty members with voice problems had significantly higher attitudes toward teaching as a cause of VD, and they had significantly more absences due to sickness than those without PVDs. These findings confirm previous studies showing that absenteeism is significantly more common among those with VD than among their healthy counterparts [21,22].

The analysis of sociodemographic factors and teaching characteristics revealed interesting findings. Previous studies have suggested that the risk of having a voice disorder increases as age increases, especially among individuals in their 40s and 50s [23,24]. However, the present study revealed no association between age and VD, which is consistent with the studies conducted by Chen et al. [25], Alva et al. [21], and Higgins and Smith [20]. In terms of the association between VDs and teaching characteristics, the study revealed that despite having a relatively higher risk of voice problems, the average number of hours spent teaching did not have a significant association with VDs. This could be due to the method used in the present study to calculate the hours, which was based on the average number of teaching hours per week rather than the number of hours or lectures per day. A recent study reported a significant correlation between the duration of consecutive hours spent teaching and the incidence of VDs [26]. Moreover, the present study also found that the number of students in a class did not contribute to the development of VDs, which is consistent with the results reported by Alva et al. [21], Devadas et al. [27], and Feng et al. [26].

Several factors have been suggested as risks for VDs, including health status, daily lifestyle, and medications. In terms of daily lifestyle, smoking showed a trend in the expected direction in this study, but the association was insignificant. Caffeine consumption had the only significant association with PVDs: voice disorders were 29 times more common among staff consuming more than six cups of caffeinated coffee and tea daily. This result reflects those of Azari et al.’s systematic review, which also found that caffeine consumption was the most frequently reported factor among teaching faculty [11]. This is anticipated, as caffeine consumption can induce dehydration and thus have a detrimental effect on phonation [28].

Additionally, several health conditions were identified in the present study as significant risk factors for PVDs, including reflux/heartburn, allergies, hearing impairment, frequent common colds, depression, and fatigue. Reflux/heartburn, allergies, and frequent common colds were reported by 44.1%, 37.9%, and 29.4% of the PVD group, respectively, and by 22.3%, 20.1%, and 16.5% of the NPVD group, respectively. Exposure to dust, fans, and air-conditioning are among the factors that can contribute to an allergy [29,30,31,32]. Having acid reflux was significantly associated with the PVD group. Similar findings have been reported in the literature, indicating a strong association between experiencing voice problems and reflux [19,27,33,34,35]. The voice can be affected due to the gastric contents moving up to the larynx, which might cause swelling and irritation of the laryngeal mucosa, and from this, causing voice symptoms to appear [34,36,37].

Various studies have reported psychological phenomena, including stress, fatigue (burnout), and depression, as major risk factors for the development of VDs [38,39,40,41]. Under stressful conditions, vocal folds may lose their ability to move precisely; therefore, increased tension, pain, and discomfort in the laryngeal muscles can cause vocal symptoms [3,42]. Many work-related factors can generate psychological distress among teaching faculty, including the use of technology, the organizational environment, the physical workspace, time pressures, hierarchical relations between colleagues and superiors, the availability of group support, inadequate schedules, psychological and teaching demands, and excess work hours [43,44]. Apart from work-related conditions, research indicates that female university teachers are more prone to psychological distress due to several factors, including mental overload and difficulties with their work–life balance [44,45,46,47]. The significant association between PVDs and hearing impairment found in the present study is in line with several studies (e.g., [48,49,50]).

Some medications, including antihistamines, can harm the voice due to endolaryngeal changes, resulting in voice changes [24]. Most of the vocal changes induced by these medications are transient and reversible, but some may be permanent [51]. In the present study, allergy medications had a significant association with PVDs, which is consistent with other reports [20]. Although medications used to treat gastroesophageal reflux disease (GERD) are successful in terms of removing symptoms related to GERD, including hoarseness [52,53], the present study showed a significant association between GERD medications and PVDs. The reason for this finding is unclear; however, Thompson [51] reported an adverse effect of anti-reflux medications on the voice. It could also be speculated that voice problems induced by anti-reflux medication are manifested as a dry throat, not hoarseness, as dryness of the mouth and throat was reported in this study as the most frequent symptom by the faculty with PVDs. However, the nature of the association remains unclear. The significant association reported in the current study between PVDs and dryness is similar to previous studies [6,7,19], in which dryness was reported to increase the thickness of the vocal fold mucus, stiffness, and friction of the vocal folds. Several possible reasons for dryness exist, including inadequate hydration, increased transglottic airflow, and having an open mouth during teaching activities [54]. Furthermore, exposure to dry environments, as in the city of Riyadh, together with the use of antihistamines [55] dietary changes [56], and mouth breathing [57] during times of nasal or sinus congestion, and the increased likelihood of upper respiratory tract infections [58] may explain the increased incidence of throat dryness.

Moving on to the faculty members’ quality of life, according to Ma and Yiu [59], quality of life refers to individuals’ functioning and participation in daily activities. The present study grouped QoL into four main domains: physical health, psychological health, social relations, and environment. Comparing the QoL between the two groups, the faculty members with VDs had significantly worse QoL in all four domains than those with NPVDs (*p* < 0.001). The reduced QoL might be related to different factors; one factor is the effect of GERD. Different studies report that voice-related symptoms such as hoarseness, chronic cough, and throat clearing lead to negative physical, emotional, and psychological health consequences. Thus, all of these symptoms lead to lower self-esteem, increased relationship fatigue, fatigue, frustration, and higher stress levels, all of which affect QoL [60,61]. Faculty members are also at increased risk of vocal strain due to their job demands, which can potentially limit their professional performance. This limitation may manifest as absences, reduced work performance, frustration, and even a desire to switch careers [62].

This study has limitations that should be considered. First, the study included only female faculty members at one university and thus cannot be generalized to other universities. Additionally, rather than using an assessment procedure that would have permitted the identification of actual cases with or without PVDs, the vocal complaint variable was evaluated based only on self-reporting. Therefore, the possibilities of measurement bias and memory bias cannot be completely ruled out. It should be noted that auditory assessments are considered the gold standard to identify voice disorders and provide immediate measures of voice quality severity; therefore, future research should consider this factor to support the conclusions reached in the current study [63,64,65].

## 5. Conclusions

Voice disorders can stem from several factors identified in the present study, including health conditions (reflux/heartburn, allergies, hearing impairment, frequent common colds, depression, and fatigue), medications (e.g., antihistamines and anti-reflux medicine), and daily lifestyles (e.g., caffeine consumption). Although faculty members with voice problems believed that teaching caused their perceived voice disorders, the findings of this study revealed that teaching characteristics did not significantly contribute to voice disorders. Moreover, demographic factors that were suspected to increase voice problems did not contribute significantly to the prevalence of perceived voice disorders. The study also found that faculty members with voice problems had significantly poorer QoL in all four domains. Thus, having identified several risk factors for developing voice problems, it is clear that teaching faculty need to be educated on preventative vocal measures and increase their awareness of voice care and the risk factors that can affect their voice. Techniques on how to relax the vocal tract when it is tense during stressful situations could be demonstrated for the faculty. Moreover, we recommend further studies using objective clinical examination to determine voice disorders and design rehabilitation programs for those in need. 

## Figures and Tables

**Figure 1 jpm-13-01568-f001:**
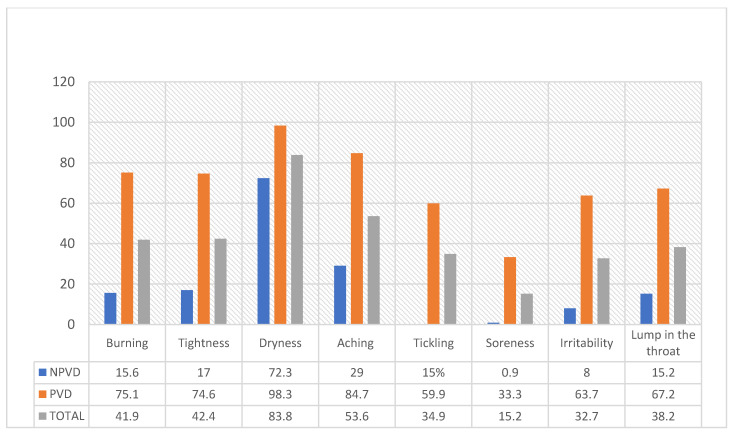
Frequency of perceived voice-related symptoms experienced during or after teaching in both study groups (N = 401).

**Figure 2 jpm-13-01568-f002:**
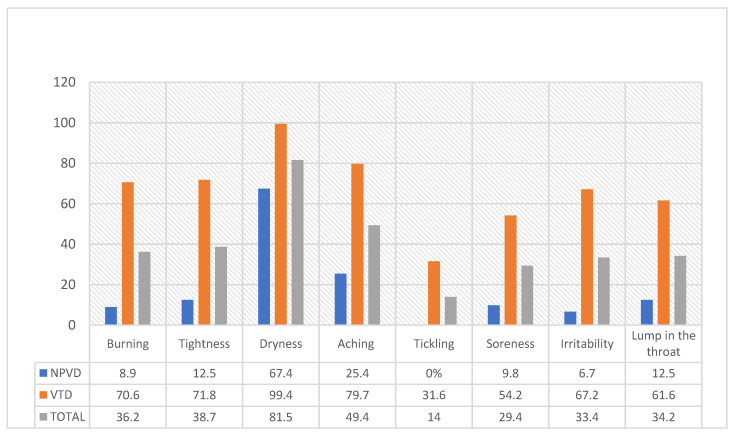
Severity of voice-related symptoms felt during or after teaching in both study groups (N = 401).

**Table 1 jpm-13-01568-t001:** Relationship between perceived voice disorders and the participants’ sociodemographic and professional characteristics (N = 401).

Characteristic		Study Group		Total n (%)	$\chi^{2}$ Test	*p*-Value	OR (95% CI)
		NPVD n (%)	PVD n (%)				
Age group (years)	≤29	95 (42.4)	79 (44.6)	174 (43.4)	0.272	0.87	1
	30–49	82 (36.6)	64 (36.2)	146 (36.4)			0.94 (0.60–1.46)
	≥50	47 (21.0)	34 (19.2)	81 (20.2)			0.87 (0.51–1.48)
Marital status	Not married ^a^	48 (21.4)	46 (26.0)	94 (23.4)	1.146	0.17	1
	Married	176 (78.6)	131 (74.0)	307 (76.6)			0.78 (0.489–1.234)
Number of children	≤2	101 (45.1)	91 (51.4)	192 (47.9)	1.58	0.12	1
	≥3	123 (54.9)	86 (48.6)	209 (52.1)			0.78 (0.52–1.15)
Academic rank	Professor	23 (10.3)	15 (8.5)	38 (9.5)	3.93	0.56	1
	Associate Professor	47 (21.0)	28 (15.8)	75 (18.7)			0.91 (0.41–2.04)
	Assistant Professor	74 (33.0)	63 (35.6)	137 (34.2)			0.75 (0.35–1.59)
	Lecturer	57 (25.4)	44 (24.9)	101 (25.2)			1.18 (0.55–2.53)
	Demonstrator	14 (6.2)	16 (9.0)	30 (7.5)			1.75 (0.67–4.62)
	Language teacher	9 (4.0)	11 (6.2)	20 (5.0)			1.87 (0.63–5.60)
Colleges **	Humanities	96 (42.9)	64 (36.2)	160 (39.9)	1.9	0.75	1
	Sciences	53 (23.7)	48 (27.1)	101 (25.2)			1.36 (0.82–2.25)
	Health Science	54 (24.1)	47 (26.6)	101 (25.2)			1.31 (0.790–2.16)
	Deanships and Institutes	16 (7.1)	14 (7.9)	30 (7.5)			1.30 (0.59–2.87)
	Applied Colleges	5 (2.2)	4 (2.3)	9 (2.2)			1.2 (0.31–4.64)
Total		224 (55.9)	177 (44.1)	401 (100.0)			

** Fisher’s exact test was used; ^a^ includes single, divorced, and widowed; NPVD: no perceived voice disorder; PVD: perceived voice disorder.

**Table 2 jpm-13-01568-t002:** Relationship between perceived voice disorders and work-related characteristics (N = 401).

Work-Related Characteristic		Study Group		Total n (%)	$\chi^{2}$ Test	*p*-Value	OR (95% CI)
		NPVD n (%)	PVD n (%)				
Teaching experience	<5 years	30 (13.4)	29 (16.4)	59 (14.7)	4.48	0.21	1
	5–9 years	55 (24.6)	30 (16.9)	85 (21.2)			0.56 (0.29–1.11)
	10–14 years	58 (25.9)	56 (31.6)	114 (28.4)			0.99 (0.53–1.87)
	≥15 years	81 (36.2)	62 (35.0)	143 (35.7)			0.79 (0.43–1.46)
Administrative positions with teaching	No	132 (58.9)	94 (53.1)	226 (56.4)	1.36	0.24	1
	Yes	92 (41.1)	83 (46.9)	175 (43.6)			1.267 (0.85–1.89)
Average number of teaching hours (hours/week)	3–5	34 (15.2)	22 (12.4)	56 (14.0)	5.75	0.22	1
	6–8	34 (15.2)	27 (15.3)	61 (15.2)			1.23 (0.59–2.56)
	9–11	56 (25.0)	31 (17.5)	87 (21.7)			0.86 (0.43–1.71)
	12–14	62 (27.7)	55 (31.1)	117 (29.2)			1.37 (0.72–2.62)
	≥15	38 (17.0)	42 (23.7)	80 (20.0)			1.71 (0.854–3.42)
Class size (average number of students/class)	10–20	30 (13.4)	22 (12.4)	521 (3.0)	7.14	0.13	1
	21–30	71 (31.7)	44 (24.9)	115 (28.7)			0.85 (0.43–1.65)
	31–40	61 (27.2)	40 (22.6)	101 (25.2)			0.89 (0.45–1.76)
	41–50	21 (9.4)	24 (13.6)	45 (11.2)			1.56 (0.698–3.48)
	≥51	41 (18.3)	47 (26.6)	88 (21.9)			1.60 (0.78–3.12)
Remote teaching	No	175 (78.1)	137 (77.4)	312 (77.8)	0.03	0.86	1
	Yes	49 (21.9)	40 (22.6)	89 (22.2)			1.04 (0.65–1.68)
Total		224 (55.9)	177 (44.1)	401 (100.0)			

NPVD: no perceived voice disorder; PVD: perceived voice disorder.

**Table 3 jpm-13-01568-t003:** Relationship between perceived voice disorders and health conditions (N = 401).

Characteristic		Study Group		Total n (%)	$\chi^{2}$ Test	*p*-Value	OR (95% CI)
		NPVD n (%)	PVD n (%)				
Thyroid gland problems	No	187 (83.5)	153 (86.4)	340 (84.8)	0.67	0.41	1
	Yes	37 (16.5)	24 (13.6)	61 (15.2)			0.79 (0.46–1.38)
Reflux/ Heartburn	No	174 (77.7)	99 (55.9)	273 (68.1)	21.5	<0.001 *	1
	Yes	50 (22.3)	78 (44.1)	128 (31.9)			2.74 (1.78–4.23)
Asthma	No	207 (92.4)	158 (89.3)	365 (91.0)	1.20	0.27	1
	Yes	17 (7.6)	19 (10.7)	36 (9.0)			1.46 (0.74–2.91)
Fever	No	216 (96.4)	170 (96.0)	386 (96.3)	0.04	0.84	1
	Yes	8 (3.6)	7 (4.0)	15 (3.7)			1.11 (0.39–3.13)
Allergies	No	179 (79.9)	110 (62.1)	289 (72.1)	15.5	<0.001 *	1
	Yes	45 (20.1)	67 (37.9)	112 (27.9)			2.42 (1.55–3.76)
Hearing impairment	No	216 (96.4)	155 (87.6)	371 (92.5)	11.21	0.002 *	1
	Yes	8 (3.6)	22 (12.4)	30 (7.5)			3.83 (1.66–8.83)
Frequent common cold	No	187 (83.5)	125 (70.6)	312 (77.8)	9.47	0.002 *	1
	Yes	37 (16.5)	52 (29.4)	89 (22.2)			2.10 (1.30–3.39)
Depression	No	205 (91.5)	146 (82.5)	351 (87.5)	7.03	0.007 *	1
	Yes	19 (8.5)	31 (17.5)	50 (12.5)			2.29 (1.25–4.21)
Fatigue	No	112 (50.0)	39 (22.0)	151 (37.7)	32.9	<0.001 *	1
	Yes	112 (50.0)	138 (78.0)	250 (62.3)			3.54 (2.28–5.50)
Total		224 (55.9)	177 (44.1)	401 (100.0)			

* *p*-value is statistically significant ≤ 0.05, NPVD: No Perceived Voice Disorder; PVD: Perceived Voice Disorder.

**Table 4 jpm-13-01568-t004:** Relationship between perceived voice problems and the treatment of health conditions (N = 401).

Characteristic		Study Group		Total n (%)	$\chi^{2}$ Test	*p*-Value	OR (95% CI)
		NPVD n (%)	PVD n (%)				
Thyroid gland problems	No	89 (39.7)	77 (43.5)	166 (41.4)	0.66	0.72	1
	Yes	29 (12.9)	20 (11.3)	49 (12.2)			0.73 (0.42–1.52)
	NA	106 (47.3)	80 (45.2)	186 (46.4)			0.78 (0.64–1.08)
Reflux/ Heartburn	No	95 (42.4)	74 (41.8)	169 (42.1)	13.86	0.001 *	0.87 (0.57–1.33)
	Yes	24 (10.7)	42 (23.7)	66 (16.5)			2.25 (1.25–4.04)
	NA	105 (46.9)	61 (34.5)	166 (41.4)			0.75 (0.48–1.16)
Asthma	No	92 (41.1)	73 (41.2)	165 (41.1)	2.25	0.33	1
	Yes	15 (6.7)	19 (10.7)	34 (8.5)			1.59 (0.76–3.36)
	NA	117 (52.2)	85 (48.0)	202 (50.4)			0.92 (0.60–1.39)
Fever	No	95 (42.4)	80 (45.2)	175 (43.6)	0.57	0.75	1
	Yes	6 (2.7)	6 (3.4)	12 (3.0)			1.19 (0.367–3.83)
	NA	123 (54.9)	91 (51.4)	214 (53.4)			0.88 (0.59–1.31)
Allergies	No	98 (43.8)	68 (38.4)	166 (41.4)	12.13	0.002 *	1
	Yes	26 (11.6)	44 (24.9)	70 (17.5)			2.44 (1.37–4.34)
	NA	100 (44.6)	65 (36.7)	165 (41.1)			0.94 (0.60–1.45)
Hearing impairment	No	98 (43.8)	92 (52.0)	190 (47.4)	3.35	0.19	1
	Yes	1 (0.4)	1 (0.0)	2 (0.4)			0.9 (0.0717.28)
	NA	125 (55.8)	85 (48.0)	210 (52.4)			0.72 (0.49–1.08)
Frequent common cold	No	100 (44.6)	77 (43.5)	177 (44.1)	4.73	0.09	1
	Yes	22 (9.8)	30 (16.9)	52 (13.0)			1.78 (0.95–3.31)
	NA	102 (45.5)	70 (39.5)	172 (42.9)			0.89 (0.58–1.36)
Depression	No	97 (43.3)	83 (46.9)	180 (44.9)	3.23	0.19	1
	Yes	9 (4.0)	13 (7.3)	22 (5.5)			1.69 (0.694.15)
	NA	118 (52.7)	81 (45.8)	199 (49.6)			0.80 (0.53–1.21)
Fatigue	No	148 (66.1)	117 (66.1)	265 (66.1)	20.53	<0.001 *	1
	Yes	9 (4.0)	28 (15.8)	37 (9.2)			3.9 (1.79–8.66)
	NA	67 (29.9)	32 (18.1)	99 (24.7)			0.60 (0.37–0.98)
Total		224 (55.9)	177 (44.1)	401 (100.0)			

* *p*-value is statistically significant ≤ 0.05, NA: not applicable; NPVD: no perceived voice disorder; PVD: perceived voice disorder.

**Table 5 jpm-13-01568-t005:** Relationship between perceived voice disorders and lifestyle (N = 401).

Characteristic		Study Group		Total n (%)	$\chi^{2}\;\mathrm{Test}$	*p*-Value	OR (95% CI)
		NPVD n (%)	PVD n (%)				
Smoking status	Never smoked	216 (96.4)	169 (95.5)	385 (96.0)	1.25	0.54	1
	Former smoker	5 (2.2)	3 (1.7)	8 (2.0)			0.77 (0.18–3.25)
	Current smoker	3 (1.3)	5 (2.8)	8 (2.0)			2.13 (0.50–9.04)
Water drunk/day	≤3 cups	134 (59.8)	115 (65.0)	15 (63.1)	2.67	0.45	1
	4–6 cups	68 (30.4)	49 (27.7)	117 (29.2)			0.84 (0.54–1.31)
	>6 cups	22 (9.8)	13 (7.3)	35 (8.7)			1.16 (0.55–2.45)
Caffeinated coffee and tea/day **	≤3 cups	211 (94.2)	165 (94.2)	376 (93.7)	0.632	<0.001 *	1
	4–6 cups	11 (4.9)	9 (5.1)	20 (5.0)			1.05 (0.42–2.58)
	>6 cups	2 (0.9)	3 (1.7)	5 (1.2)			28.77 (4.35–190.28)
Decaffeinated coffee and tea/day **	≤3 cups	180 (80.4)	146 (82.5)	326 (81.3)	0.29	0.59	1
	4–6 cups	44 (19.6)	31 (17.5)	75 (18.7)			0.87 (0.52–1.45)
Practice other hobbies using their voice	No	186 (83.0)	135 (76.3)	321 (80.0)	2.83	0.09	1
	Yes	38 (17.0)	42 (23.7)	80 (20.0)			1.52 (0.93–2.49)
Total		224 (55.9)	177 (44.1)	401 (100.0)			

* *p*-value is statistically significant ≤ 0.05; ** Fisher’s exact test was used; NPVD: no perceived voice disorder; PVD: perceived voice disorder.

**Table 6 jpm-13-01568-t006:** Relationship between perceived voice disorders, and WHOQOL-BREF and its subscales (N = 401).

		Study Group			
Variable	Overall (N = 401) M ± SD	NPVD (*n* = 224) M ± SD	PVD (*n* = 177) M ± SD	T	*p*-Value
Overall QoL Mdn (IQR)	4 (1)	4 (1)	4 (1)	1432.5 ^U^ (z = −5.10)	<0.001 *
Overall health satisfaction Mdn (IQR)	4 (2)	4 (2)	4 (1)	1470.3 ^U^ (z = −4.81)	<0.001 *
Physical health M ± SD	15.09 ± 2.83	15.90 ± 2.47	14.02 ± 2.92	6.12	<0.001 *
Psychological health M ± SD	14.31 ± 2.69	14.90 ± 2.52	13.51 ± 2.72	4.65	<0.001 *
Social relation M ± SD	14.61 ± 3.66	15.44 ± 3.29	13.49 ± 3.86	4.75	<0.001 *
Environment M ± SD	14.30 ± 2.54	14.97 ± 2.38	13.40 ± 2.49	5.55	<0.001 *
WHOQOL-BREF M ± SD	58.32 ± 10.09	61.22 ± 8.91	54.43 ±10.32	6.17	<0.001 *

* *p*-value is statistically significant ≤ 0.05, NPVD: no perceived voice disorder; PVD: perceived voice disorder; ^U^ = Mann–Whitney test; T = independent *t*-test; IQR: interquartile range; M: mean; Mdn: Median; SD: standard deviation; WHOQOL-BREF: World Health Organization Quality of Life-Brief.

## Data Availability

Not applicable.

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
