# Peer review of "Measuring Perceived Voice Disorders and Quality of Life among Female University Teaching Faculty"

_jpm, 2023, doi:10.3390/jpm13111568_

Round 1
Reviewer 1 Report
Comments and Suggestions for Authors
The draft is well conceived and supported by statistical analisys. The authors described the limits of the study .Conclusions are clear and supported by results. I suggest publication.
Author Response
Much appreciated. Thanks for your time
Reviewer 2 Report
Comments and Suggestions for Authors
Professions that require intense voice use may put the person at risk of voice disorders. There are many similar studies. However, perceived voice impairment indicates an awareness of faculty members in terms of quality of life in terms of this study.
Comments on the Quality of English LanguageEnough,
Author Response
Much appreciated. Thanks for your comment
Reviewer 3 Report
Comments and Suggestions for Authors
Thank you for the opportunity to review the article entitled “Measuring perceived Voice disorders and Quality of Life among Female University Teaching Faculty”.
This study aims to measure the perceived voice problems among 401 female university faculty, identify potential risk factors, and determine the effect of voice problems on their quality of life. Results indicated that faculty members with VD have poorer health conditions, highlighting the need for education on preventative vocal measures and awareness of voice care. Here are a few minor suggestions.
Specific comments:
- I totally agree with the authors that voice disorders are common in the teaching profession, although they are not classified as elite vocal performers. A recent study compared subjective voice activity limitation and participation restriction according to the level of professional voice use when controlling for benign vocal disease, gender, and severity using the VAPP questionnaire. Results showed that elite vocal performer only had more subjective voice problems related to their jobs than non-professionals, while professional voice users such as teachers and professors also complained of greater voice problems related to social communication and job. I suppose these results also support the need for this study.
https://doi.org/10.12963/csd.19649
- It is unclear what different results Table 3 and Table 4 show, so additional explanation is needed.
- As you pointed out, auditory-perceptual evaluation by certified SLPs would have supported the conclusion of the current study. It is truly a limitation because patient with more severe dysphonia report more subjective vocal complaint. I think this should be emphasized more.
Comments on the Quality of English LanguageNone.
Author Response
Thank you very much for your careful reading of our text.
We very much appreciate your valuable comments and suggestions, which have been very helpful in improving the manuscript. All the comments we received on this study have been taken into account in improving the quality of the article, and we present our reply to each of them. We believe that your suggestions have been very helpful in improving the manuscript.
Please find the information requested and the clarification in explaining the tables below.
|
Reviewer 3 |
Authors’ reply |
|
Thank you for the opportunity to review the article entitled “Measuring perceived Voice disorders and Quality of Life among Female University Teaching Faculty”. This study aims to measure the perceived voice problems among 401 female university faculty, identify potential risk factors, and determine the effect of voice problems on their quality of life. Results indicated that faculty members with VD have poorer health conditions, highlighting the need for education on preventative vocal measures and awareness of voice care. Here are a few minor suggestions.
|
Thank you very much for your valuable comments.
|
|
Specific comments: - I totally agree with the authors that voice disorders are common in the teaching profession, although they are not classified as elite vocal performers. A recent study compared subjective voice activity limitation and participation restriction according to the level of professional voice use when controlling for benign vocal disease, gender, and severity using the VAPP questionnaire. Results showed that elite vocal performer only had more subjective voice problems related to their jobs than non-professionals, while professional voice users such as teachers and professors also complained of greater voice problems related to social communication and job. I suppose these results also support the need for this study. https://doi.org/10.12963/csd.19649
|
Thank you for sharing the reference. It was added to the discussion as follows: “An example is a study that compared subjective voice activity limitation and participation restriction based on the level of professional voice use after controlling for benign vocal disease, gender, and severity using the VAPP questionnaire. According to the study results, elite vocal performers experienced more subjective voice problems related to their jobs than non-professionals, while professional voice users, including teachers and professors, also reported more significant voice problems related to social communication and employment” |
|
- It is unclear what different results Table 3 and Table 4 show, so additional explanation is needed.
|
Thank you very much for your valuable comments. Additional explanation is added as the following: The results for measuring the relationship between perceived voice disorders and health conditions and if it is treated or not are displayed in tables 3 and 4. Whereas, the relationship between perceived voice disorders and health conditions revealed statistically significant associations between perceived voice problems and most of the investigated health conditions (P-values ≤ .05). Voice problems were almost four times significantly higher among those suffering from hearing impairment and fatigue (OR 3.83, 95% CI 1.66–8.83, P = .002 and OR 3.54, 95% CI 2.28–35.50, P < .001, respectively). Similarly, voice problems were nearly three times higher among the staff suffering from reflux (OR 2.74, 95% CI 1.78–4.23, P < .001) and around twice higher among the staff suffering from allergies (OR 2.42, 95% CI 1.55–3.76), depression (OR 2.29, 95% CI 1.25–4.21, P= .007), and frequent common colds (OR 2.10, 95% CI 1.30–3.39, P < .001) (Table 3). On the other hand, on assessing the relationship between perceived voice problems and the treatment of the previously mentioned health conditions in table 3, the results revealed only 2 of the treated health conditions which are namely allergies and reflux were statistically significant associated with their perceived voice problems (P-values ≤ .05). Voice disorders were nearly two and a half times significantly higher among those receiving treatment for allergies and reflux (OR 2.44, 95% CI 1.37–4.34, P = .002 and OR 1.59, 95% CI 0.76–3.36, P = .001, respectively) (Table 4). |
|
- As you pointed out, auditory-perceptual evaluation by certified SLPs would have supported the conclusion of the current study. It is truly a limitation because patient with more severe dysphonia report more subjective vocal complaint. I think this should be emphasized more.
|
This paragraph is now added: “It should be noted that auditory assessments are considered the gold standard to identify voice disorders and provide immediate measures of voice quality severity; therefore, future research should consider this factor to support the conclusions reached in the current study”
|